# ATRX/DAXX: Guarding the Genome against the Hazards of ALT

**DOI:** 10.3390/genes14040790

**Published:** 2023-03-24

**Authors:** Sarah F. Clatterbuck Soper, Paul S. Meltzer

**Affiliations:** Genetics Branch, Center for Cancer Research, National Cancer Institute, Bethesda, MD 20892, USA

**Keywords:** ATRX, DAXX, Alternative Lengthening of Telomeres

## Abstract

Proliferating cells must enact a telomere maintenance mechanism to ensure genomic stability. In a subset of tumors, telomeres are maintained not by telomerase, but through a homologous recombination-based mechanism termed Alternative Lengthening of Telomeres or ALT. The ALT process is linked to mutations in the ATRX/DAXX/H3.3 histone chaperone complex. This complex is responsible for depositing non-replicative histone variant H3.3 at pericentric and telomeric heterochromatin but has also been found to have roles in ameliorating replication in repeat sequences and in promoting DNA repair. In this review, we will discuss ways in which ATRX/DAXX helps to protect the genome, and how loss of this complex allows ALT to take hold.

## 1. Introduction

To counteract the erosion of DNA ends due to the end-replication problem, organisms with linear chromosomes must enact a program of telomere maintenance. In mammalian cells, telomeres are lengthened through the action of the ribonucleoprotein telomerase, which synthesizes new DNA repeats from an RNA template. Telomerase is not expressed in most somatic cells, however, so in the process of immortalization cancer cells must acquire a telomere maintenance program. Most tumors reactivate the expression of telomerase, but 10–15% of tumors lengthen telomeres through a DNA-templated process known as Alternative Lengthening of Telomeres or ALT [1,2,3,4].

ALT is strongly correlated with loss of ATRX or DAXX [5,6,7], which together form a complex that deposits non-canonical histone variant H3.3 at pericentromeric and telomeric heterochromatin [8,9,10]. Though ALT-related ATRX or DAXX mutations are often truncating nonsense mutations or deletions, protein loss has been observed without obvious coding sequence changes, implying loss of expression due to alterations in promoters or splicing, or due to epigenetic changes [6]. Notably, ATRX and DAXX loss are mutually exclusive. ALT is so highly correlated with loss of ATRX or DAXX that mutations leading to truncation of these proteins have been considered synonymous with ALT status, though using this metric as a reliable diagnostic of ALT has also been controversial [11,12,13]. ATRX/DAXX mutations associated with ALT telomere maintenance are most frequently observed in liposarcomas, adult gliomas, pancreatic neuro-endocrine tumors, and osteosarcomas [11]. ALT is rarely observed in carcinomas, or tumors arising from highly proliferative tissue, perhaps due to their predisposition to reactivate telomerase. Many ALT-prone tumors have poor prognoses, so a better understanding of the role of ATRX and DAXX mutations in ALT and in tumorigenesis generally is desirable to open the door to new targeted therapies.

In this review, we will explore the normal function of ATRX and DAXX in safe-guarding the genome, and describe recent advances in the understanding of how mutations in the ATRX/DAXX complex promote acquisition of the ALT phenotype.

## 2. The ATRX/DAXX Complex

ATRX is named for its causal role in ATR-X syndrome (α-thalassemia with mental impairment, X-linked), an X-linked disorder characterized by developmental delays, urogenital abnormalities, distinctive craniofacial features, and α-thalassemia caused by insufficient α-globin expression [14]. Because of the central role of decreased α-globin mRNA expression in the ATR-X phenotype, research on ATRX initially focused on its potential as a transcriptional regulator [14,15,16]. In fact, ATRX in concert with DAXX play wide-ranging roles in maintaining chromatin and reckoning with problematic DNA repeat sequences, with downstream effects on gene expression that have critical impacts in development.

The *ATRX* gene is found on chromosome Xq [17]. It expresses a full-length protein of 280 kDa as well as shorter alternatively spliced isoforms [18,19]. ATRX is largely disordered [20] but features two ordered domains, an N-terminal ADD (ATRX-DNMT3-DNMT3L) domain and a C-terminal helicase-like domain (Figure 1a,b). The helicase-like domain is related to those of the SNF2 family of helicase-related proteins [17,19], a protein family containing DNA translocases and chromatin remodelers. The particular SNF2 domain variant found in ATRX places it into the Rad54-like sub-family, other members of which have roles in altering local DNA torsion, strand-exchange and directing DNA methylation [21]. Notably, RAD54-like proteins are not known to be efficient in terms of canonical chromatin remodeling function [22].

In addition to the SNF2 domain, ATRX features an N-terminal ADD domain, a motif related to the Plant Homeodomain (PHD) [25]. This domain acts as an atypical H3K9me3 reader module [26], allowing the targeting of ATRX to heterochromatin [27]. About half of mutations observed in ATR-X syndrome cluster in the ADD domain, while a third are found in the helicase motifs [28]; some disease-associated mutations impair ATRX protein localization within the nucleus [29]. ATR-X associated mutations are mostly missense variants and are expected to be hypomorphs [28], as complete loss of ATRX is embryonic lethal in mice [30]. Interestingly, there is some indication that ATR-X patients, especially those with nonsense mutations in the helicase-like domain, may have a predisposition to osteosarcoma [31,32,33]. Unlike constitutive mutations observed in ATR-X syndrome, somatic *ATRX* mutations found in cancers are usually truncating non-sense mutations that are expected to completely abrogate ATRX activity [11].

ATRX forms a complex with the H3.3 specific histone chaperone DAXX [9,22,34]. DAXX is a predicted 81 kDa protein encoded from a gene located at 6p in humans. It was originally described as an effector of Fas mediated apoptosis and proposed to directly contact Fas [35]. In retrospect, the protein interaction data that initially pointed to this role may have been artifactual [35,36], as Fas is a cell surface receptor and endogenous DAXX localizes to the nucleus [37,38]. In another example of early misconceptions about the function of DAXX, it was initially characterized as having a pro-apoptotic activity, but the DAXX knock-out mouse phenotype is embryonic lethal with extensive apoptosis [39]. We now recognize that DAXX is a multifunctional protein with numerous interaction partners, though its function in complex with ATRX is the one known to be closely tied to ALT.

Though widely expressed in human cells and conserved through the animal kingdom, DAXX has no known paralogs [40]. Like ATRX, the DAXX protein is predicted to be mostly disordered (Figure 1a,b) [20]. The two ordered regions of DAXX include the N-terminal 4-helix-bundle (4HB) that interacts with a conserved helical stretch of ATRX [41], as well as the histone binding domain. In fact, the DAXX histone binding domain is only ordered contingent upon histone binding; it behaves as an unfolded protein in the absence of H3.3 [42]. DAXX and H3.3 co-fold upon binding, stabilizing both the H3.3/H4 heterodimer and DAXX. DAXX enfolds the H3.3/H4 heterodimer [43], making contact with the G90 residue that is unique to the H3.3 variant to ensure specificity [44]. In the absence of H3.3 binding, the entire DAXX protein becomes unstable [41].

ATRX and DAXX interact through the DAXX 4HB motif, in contact with the DAXX binding domain of ATRX—a short, conserved motif in the central part of the ATRX protein (Figure 1a) [34,41]. Interruption of the DAXX-ATRX interface in mouse embryonic stem cells (mESCs) results in gene expression profile changes that mirror the genetic knock-out of *Daxx,* implying that DAXX requires ATRX to be functional for gene regulation. Indeed, cancer-associated missense mutations in DAXX cluster in the 4HB ATRX-binding interface in addition to the histone binding domain, indicating the importance of this complex formation in suppression of tumorigenesis [41].

## 3. Localization and Function of ATRX/DAXX

ATRX/DAXX are known predominantly for their role in deposition of the non-canonical histone variant H3.3 in pericentromeric and telomeric heterochromatin. Histone variant H3.3 is unlike the canonical H3 variants in that its synthesis and deposition are replication-independent—that is, not tied to new DNA synthesis in S-phase [45]. For this reason H3.3 can be characterized as a “replacement” histone, which can restore chromatin status in regions of the genome where histones may have been lost or displaced outside of S-phase [46]. Though DAXX is responsible for directly binding H3.3, both complex partners are required for deposition [9]. ATRX/DAXX are not the only histone chaperones for variant H3.3. While ATRX/DAXX deposit H3.3 in heterochromatin [8,9,10,47], another histone chaperone, HIRA deposits H3.3 in actively transcribed regions of the genome, including promoters, enhancers and gene bodies [8,48,49]. In the absence of ATRX/DAXX, HIRA can compensate for H3.3 deposition to telomeric heterochromatin, as well.

The ATRX/DAXX complex is enriched at PML nuclear bodies [22,50], historically known as ND10. PML bodies are discrete, spherical, nuclear foci scaffolded by PML protein. PML bodies range in size from about 0.1–1 µm, and the number of PML bodies per cell ranges from 5–30, depending on the cell type and status. Unlike other nuclear bodies, PML bodies are not known to have a constitutive nucleic acid component, but PML protein forms a shell around an inner core containing a multitude of protein factors (reviewed in [51]). PML bodies are organized through SUMO (small ubiquitin-related modifier) interactions with SIM (SUMO interacting motif) domains. PML protein itself is SUMOylated, and PML bodies enrich for proteins containing SIM domains [52]. Thus, DAXX is recruited to PML bodies via its SIM domains [53,54], and ATRX localization to PML bodies depends on its interaction with DAXX [34]. PML bodies appear to serve as a depot for soluble H3.3-H4 heterodimers, poised for deposition by ATRX/DAXX [55]. In the absence of PML protein, ATRX/DAXX loses its capacity to load H3.3 onto PML-associated chromatin domains, though HIRA remains functional for H3.3 deposition to these regions [56]. This finding highlights the special relationship between ATRX/DAXX and PML.

ATRX/DAXX target DNA repeat sequences, especially G-rich repeats, and sequences prone to form secondary structures. On binding, the complex plays a role in the epigenetic silencing of these sequences [57]. The G-rich TTAGGG telomere repeat is a typical ATRX target, and indeed ChIP-seq experiments reveal that telomeres bind ATRX/DAXX [8]. These observations are supported by immunofluorescence experiments in mESCs showing ATRX/DAXX localization to telomeres in a manner dependent upon H3.3 [58]. Knock-down of ATRX results in increases in telomere dysfunction induced foci (TIFs), which are evidence of telomeric DNA damage [58]. Additional ChIP-seq experiments in mESCs and human erythroid cells both reveal a preference for ATRX in binding GC-rich DNA sequences—in human cells the ATRX binding was predominantly enriched in sub-telomeres, while in mESCs ATRX targets were observed to be distributed more evenly across the genome [57].

The association with G-rich structures turns out to be fundamental to the α-thalassemia phenotype in ATR-X syndrome. The α-globin locus lies near a region of highly polymorphic, GC-rich, variable length tandem repeats known as the φζ VNTR, which contains an ATRX binding peak. When ATRX is mutated, expression of the α-like globin genes close to this region is decreased in a manner proportional to the proximity of the gene to the ATRX binding peak in the VNTR. The length of this repeat region also correlates with the severity of the ATR-X syndrome phenotype in affected individuals [57].

G-rich DNA sequences like those that recruit ATRX/DAXX are prone to the formation of G-quadruplex (G4) structures. G4s are non-B-form DNA secondary structures that can form in G-rich DNA nucleic acid sequences such as telomeres [59]. They are quite stable, and can form an impediment to transcription or DNA replication (reviewed in [60]). In vitro, ATRX binds G4s [57], though it is not competent to unwind the structures independently [61]. In neural progenitor cells lacking ATRX, small molecule mediated stabilization of G4s results in increased DNA damage and decreased cell viability [62]. Additionally, evidence suggests that ATRX/DAXX promote transcription through G-rich coding regions [63].

ATRX/DAXX aid in maintenance of the chromatin state of silenced elements, but also play roles in euchromatin. Knockout of DNA methyltransferases in mESCs results in DNA hypomethylation and derepression of tandem repetitive elements; this hypomethylated state strengthens recruitment of ATRX/DAXX to repeat elements, promoting trimethylation of H3K9 by Suv39h [64]. ATRX/DAXX also promote H3.3 deposition, H3K9 trimethylation, and silencing at imprinted alleles throughout the mouse genome [65]. Notably, H3.3 affects mouse ERV repression independently of its deposition through its stabilization of the DAXX protein, so it is unclear how crucial H3.3 deposition *per se* is to the repression of repeat element transcription. 

ATRX is a major interactor of the non-coding Telomere Repeat RNA, TERRA [66]. In mammals, TERRA is transcribed from sub-telomeric CpG islands by RNA Pol II using the C-rich strand as template, resulting in transcripts of heterogeneous length containing some sub-telomeric sequence and a variable number of TTAGGG telomere repeats [67,68,69]. TERRA features a canonical 5’ m^7^G cap, and a fraction of it is poly-adenylated [70]. TERRA transcription is increased at short telomeres and is repressed on elongated telomeres by the presence of H3K9me3 and HP1α [71]. The exact chromosomal origins of TERRA have been controversial, with some arguments that TERRA is transcribed from as few as two telomeric loci [72,73]. Current evidence indicates instead that TERRA originates from as many as 20 human sub-telomeres [69,74,75,76,77].

TERRA is functional both in *cis,* acting at the telomere where it is transcribed [78], and in *trans* from a nucleoplasmic pool of TERRA molecules [66,79,80]. In human cells, TERRA is recruited more efficiently to short telomeres than to longer telomeres [79]. While TERRA is proposed to recruit telomerase to shortened telomeres in *Saccharomyces cerevisiae* [71], this mechanism does not appear be conserved in mammals. In mammals, TERRA is observed to promote maintenance of telomeric heterochromatin through interactions with TRF2, PRC2, and SUV39H1 [77,81,82]. There is evidence that TERRA acts as an ATRX “sponge”, titrating ATRX away from DNA targets—depleting TERRA increases ATRX occupancy at rDNA, telomeres, sub-telomeres, and repetitive sequences in mESCs [66,83].

TERRA molecules can anneal at telomeric DNA sequences forming RNA-DNA hybrids known as R-loops [84]. R-loops are detectable at telomeres in wild-type human cells and are modulated through the activity of RNaseH1. Overexpression of transgenic TERRA increases R-loops and metrics of telomere fragility. When TERRA forms R-loops with telomeric sequences, it displaces the G-rich DNA strand, which is then available to form G4 structures. Unsurprisingly, R-loops and G4s are highly associated at TERRA binding sites. Consistent with this observation, knockdown of TERRA decreases genomic G4s, and knockdown of ATRX increases them [83], suggesting that TERRA antagonizes ATRX from its role in resolving G4 structures. Thus, TERRA levels may act to modulate ATRX binding to genomic targets.

R-loops can form co-transcriptionally when a nascent RNA remains annealed to the DNA template, but TERRA R-loops have been found to also form in *trans* from exogenous TERRA templates. Invasion of TERRA to telomeres in *trans* is dependent upon the recombinase RAD51 [79]. Recruitment of ATRX to telomeres has been shown to depend on transcription at the telomere and the presence of R-loops [85]. Repletion of ATRX to cells lacking it decreases R-loop levels.

In keeping with the role of ATRX in managing G4s and R-loops, both of which can impede DNA replication (reviewed in [86] and [87]), loss of ATRX contributes to an increase in replication fork stalling and DNA damage [61]. In human cells lacking ATRX, replication fork progression is impaired, replication-associated DNA damage is increased, and fork restart is defective [88]. These problems are especially prevalent in heterochromatic regions of the genome, resulting in increased activity of poly-ADP-ribose polymerase (PARP) PARP-1 to protect stalled replication forks [89]. Without the safe-guarding of the ATRX/DAXX complex, stalled replication forks are degraded by the Mre11-Rad50-Nbs1 (MRN) complex, leading to DNA breaks and genomic instability.

In addition to these roles protecting G-rich and repeat sequences, ATRX is required for resolution of sister chromatid cohesion during mitosis. Loss of ATRX results in abnormal chromosome segregation during mitosis, leading to activation of the spindle checkpoint and mitotic delays [90]. Knockdown of ATRX decreases the occupancy of cohesin at sub-telomeres [91]. This loss of cohesion at telomeres has downstream effects on DNA repair at telomeres [92]. Resolution of telomere cohesion requires the TRF1-binding poly-ADP-ribose polymerase Tankyrase 1. One model suggests that absence of ATRX leads to an excess of free histone macroH2A1.1, which binds and sequesters tankyrase 1, preventing it from resolving telomere cohesion during mitosis [93].

ATRX/DAXX also function in the repair of DNA double-stranded breaks (DSBs) by homologous recombination. In homologous recombination, the break end is first resected, then coated with Rad51 to form an invading filament that performs a homology search for the sister chromatid. Once strand invasion occurs and Rad51 is displaced, replacement DNA sequence is copied from the sister chromatid. At this point, repair can proceed through one of two sub-pathways—synthesis-dependent strand annealing (SDSA) or a double Holliday junction (dHJ), where SDSA utilizes short-patch DNA synthesis and does not result in cross-overs. A dHJ can be resolved, resulting in 50% crossover events, or dissolved through the action of BLM dissolvase, generating only non-crossover products. Recent work has demonstrated that ATRX/DAXX function to extend new DNA synthesis at DSBs, increasing the rate of long-track gene conversion and biasing repair toward the dHJ mechanism [94]. It is speculated that ATRX/DAXX deposition of H3.3 compensates for torsional strain surrounding the DNA replication complex, thus improving fork progression, and permitting long-patch synthesis. Follow-up work showed that cells using the ATRX HR sub-pathway channel HR intermediates into resolution, resulting in 50% rates of cross-over events. This outcome stands in contrast to short-patch gene conversion events in SDSA, or dissolution of intermediates by BLM, with no crossover [95].

Though the focus of this review is specifically the ATRX/DAXX complex, it is important to be aware that DAXX is estimated to be more abundant in the nucleus than ATRX [96] and is known to participate in other complexes with other binding partners. For example, DAXX was shown to function in complex with SETDB1–KAP1–HDAC1 to silence endogenous retroviruses in mouse cells [41]. There is significant evidence that DAXX can act as a transcriptional repressor [97,98], including, significantly, for p53 target promoters [99]. DAXX forms liquid-liquid phase separated assemblies with the tumor suppressor SPOP [100]. And finally, DAXX has been proposed to represent a new class of non-canonical protein folding chaperones [101]. While it seems that DAXX must bind H3.3 while participating in these all these roles due to the requirement of H3.3 for DAXX stability [41], it may be that H3.3 acts only as a passive passenger when DAXX engages in ATRX-independent functions.

## 4. ATRX/DAXX and the Suppression of ALT

ALT leverages recombination machinery to synthesize new telomere sequence. At the time of the initial discovery of ALT in mammalian cells the possibility of telomere maintenance without telomerase was already well established in other organisms, including yeast and *Drosophila* [102,103,104]. In budding yeast, it was recognized that multiple recombination pathways could compensate for a lack of telomerase activity [103]. Thus, the notion that human telomeres could be lengthened via a recombination mechanism was not difficult to imagine, and indeed it was soon found that in ALT cells lines telomere sequence was copied to other telomeres, implicating a recombination mechanism [105].

In surveys of cell lines and tumors without telomerase activity, it was observed that loss of ATRX/DAXX was a consistent feature [5,6,7]. Recent large-scale sequencing studies demonstrated that across tumor types truncation of ATRX or DAXX correlates strongly with telomere variant repeats and telomere insertions into non-telomeric regions with concurrent copy number loss at the insertion site [11]. These sequencing results highlight the risk of ALT in driving genome instability and tumorigenesis.

ALT is more common in tumors of mesenchymal origin, including osteosarcoma [5]. While ALT correlates strongly with ATRX mutations in osteosarcoma, DAXX mutations are frequently observed in pancreatic neuroendocrine tumors (PanNETs) and correlate with poorer prognosis [11,106,107,108]. Patients with multiple endocrine neoplasia-1 (MEN-1) syndrome have a predisposition to PanNETs. In these patients ATRX/DAXX was found to be intact in microadenomas but had been lost in some larger PanNETs with concurrent acquisition of ALT, implying that the acquisition of a telomere maintenance program is a late even in PanNET tumorigenesis [109]. In pediatric gliomas, mutations in H3.3 genes are also observed in addition to ATRX mutations, with strong correlation to the ALT phenotype [110].

ALT refers to a set of related mechanisms that leverage the homologous recombination machinery to achieve maintenance of telomere length in the absence of telomerase. Hallmarks of ALT telomere maintenance include long and heterogeneous telomeres [4], extra-chromosomal telomere repeat DNA (ECTRs) [111], clustering of telomere repeats at PML nuclear bodies forming ALT-associated PML Bodies (APBs) [112,113], and elevated levels of telomere sister chromatid exchange [114]. Together, these symptoms of ALT paint a picture of telomeres that have broken free of regulation. They can also serve as markers to assay for ALT activity.

ECTRs include both linear and circular DNA species and may have multiple origins. ECTRs are known to be the result of telomere trimming by XRCC3 and Nbs1 [115], but may also form from internal DNA loops (i-loops) in ALT cells [116]. Of the circular ECTRs, there exist T-circles that are double-stranded, and C-circles that are predominantly single-stranded. These C-circles can be specifically amplified through rolling-circle amplification, self-primed from a double-stranded region [117]. Detection of C-circles using this assay is a common metric of ALT activity. Notably, ECTRs can also be found in the cytoplasm in ALT cell lines, which would be expected to trigger the cGAS-STING DNA sensing pathway to trigger production of interferon b and the type I interferon response. It was found that STING expression was lost in ALT cells, but even when STING was restored to the ATRX_null_ ALT cell line U2OS it was necessary to also restore ATRX to recover functional DNA sensing [118]. This implies that a role in DNA sensing represents yet another function for the ATRX/DAXX complex.

The presence of APBs is another key indicator of ALT activity. In ALT, telomeres cluster at PML bodies forming large, intense foci [112]. APBs are the location of new DNA synthesis in ALT, and ALT does not proceed in the absence of PML protein [119,120]. As previously mentioned, PML bodies are organized through SUMO-SIM interactions, and APBs are no exception to that rule. In ALT, the SMC5/6 complex SUMO ligase MMS21, as well as the SUMO ligase PIAS4, sumoylate telomere binding proteins including TRF1 and TRF2 [121,122]. This sumoylation is required for recruitment of telomeres to APBs [121]. Curiously, synthetic APB-like condensates have been induced to form in cells to test the notion that it is the liquid properties of APBs rather than the specific proteins that drive clustering [123]. Indeed, it was possibly to induce the clustering of telomeres with PML bodies using a SIM-coupled dimerization-induction system, but these clusters were not functional for new telomere synthesis. When telomeres were tethered with PML in an ALT cell line, however, new DNA synthesis was generated at the synthetic APB [122]. Thus, the ALT mechanism requires correct signaling to engage the DNA repair machinery, not mere telomere proximity to PML.

If proximity is insufficient, how is ALT triggered *de novo*? Despite the strong correlation of ATRX/DAXX loss to ALT, ablation of ATRX is of itself insufficient to trigger the ALT phenotype [61,91,124], though upon crisis cells lacking ATRX are predisposed to activate ALT instead of telomerase [124]. One successful strategy for activating ALT *de novo* in telomerase-positive fibrosarcoma cells involved shRNA knockdown of both ATRX and DAXX plus infliction of constitutive telomere damage. This was accomplished using inducible overexpression of a TPP1 ∆OBRD fold construct to induce telomere-specific DNA damage and inhibit telomerase activity [125]. In this context, C-circle production was triggered, cells formed APBs, and telomere length was maintained, indicating adoption of ALT telomere maintenance.

ALT has also been activated through depletion of the histone H3/H4 chaperone ASF1 in human primary fibroblasts and immortalized cell lines [126]. ASF1 is an H3/H4 chaperone that delivers both canonical replication-coupled H3/H4 heterodimers to the replication fork as well as providing H3.3/H4 heterodimers to ATRX/DAXX and HIRA. The co-depletion of ASF1a and ASF1b induces C-circle production, APBs, and telomere length maintenance in absence of telomerase. This induction of ALT is relatively rapid, occurring withing 72 h of ASF1 depletion, and persists even after ASF1 protein levels rebound. These findings are phenocopied by loss of TLK (tousled-like kinase) activity. The TLKs (TLK1 and TLK2) are Ser-Thr kinases that regulate the activity of ASF1A and ASF1B. Depletion of TLKs results in replication stress and impaired *de novo* nucleosome assembly [127]. In the ALT+ U2OS osteosarcoma cell line, depletion of TLKs results in increases in markers of ALT activity, while in telomerase-expressing HeLa cells APBs and C-circles were observed when TLK1 was knocked out. Taken together, these experiments strongly suggest that provisioning of H3.3 at the replication fork is necessary for ALT suppression. Importantly, it was recently observed that HIRA can compensate for H3.3 deposition at telomeres in ALT cells, and loss of HIRA is synthetic lethal with ALT [128]. Thus, ALT seems tied to reduction of telomeric H3.3, but total elimination of H3.3 at telomeres is not tolerated.

How is ALT DNA synthesis accomplished? Though details of the ALT mechanism continue to be worked out, current evidence indicates that new DNA synthesis at ALT telomeres may proceed through at least three distinct RAD51 independent pathways (Figure 2). These pathways differ in their details but may all be considered variations of a break-induced replication (BIR) mechanism [129,130]. BIR is a repair pathway for single-ended DNA breaks that results in conservative synthesis of new DNA. Single-ended breaks can occur due to the collapse of a replication fork or can be mimicked by an eroded telomere. As in other modes of homologous recombination, BIR begins with resection at the break site to produce a single-stranded filament to engage in a homology search. From there, the separate sub-pathways diverge. In a model system with experimentally-generated DSBs at telomeres, breaks are repaired through a RAD52, SLX4 independent mechanism termed Break Induced Telomere Synthesis or BITS [129], which can extend into mitosis. RAD52 and SLX4 are both required for telomeric mitotic DNA synthesis (MiDAS), a DNA repair pathway used at DNA common fragile sites and observed at telomeres in both ALT and telomerase positive cells [130,131]. As MiDAS is relatively infrequent, most ALT telomere synthesis spontaneously arises in the G2 phase and utilizes a RAD52 dependent, SLX4 independent pathway [132]. After homology search and strand invasion, the ALT mechanism absolutely requires the action of the BLM–TOP3A–RMI (BTR) dissolvase complex to promote long-track DNA synthesis by PCNA-POLD3, then dissolve the resulting recombination intermediate without crossover [133,134].

ALT telomeres have reduced chromatin compaction, consistent with the role of ATRX/DAXX in H3.3 deposition. In addition to increased nuclease sensitivity, ALT telomeres are low in H3K9me3 and express increased levels of TERRA [6,67,68,135]. TERRA expression appears to be a critical driver of ALT in multiple ways. TERRA transcription destabilizes chromosomes and increases their replication stress. Inhibition of TERRA expression reduces ALT activity [74]. In ALT, TERRA recruits the DNA repair factor RAD51AP1, which is thought to promote formation of recombination intermediates that are important for ALT [136]. One model suggests that RAD52 and RAD51AP1 play distinct roles at telomeres, where RAD52 promotes telomeric D-loops and RAD51AP1 enables formation of R-loops with TERRA. These R-loops represent a sort of HR intermediate that can be “swapped out” in a RAD52 independent manner for an invading D-loop to enable BIR synthesis [137].

In ALT cell lines that lack fully functional ATRX or DAXX, repletion of the dysfunctional protein is sufficient to suppress ALT. When ATRX was restored to U2OS cells, C-circles were reduced, APBs diminished, and telomeres eroded, indicating that ALT was no longer active in these cells. Histone H3.3 occupancy was increased at telomeres and replication stress was alleviated. These effects were mitigated by the addition of a G4 stabilizing drug, indicating that the effect of ATRX was at least in part related to its role in reducing replication stress due to G4s [138]. In other work, wild-type DAXX was restored to the ALT + G292 osteosarcoma cell line, in which ATRX is wild type but DAXX has undergone a fusion event with the non-canonical kinesin KIFC3. In this cell line DAXX has lost the C-terminal SIM motif that targets it to PML bodies, resulting in mislocalization of DAXX and ATRX. Restoration of wild-type DAXX in G292 localizes ATRX and abrogates ALT. Notably, the endogenous DAXX in G292 is nearly full-length except for the localization motif, and it binds both ATRX and H3.3 competently, indicating that presence of ATRX/DAXX not only in the nucleus but specifically at PML bodies is essential for ALT suppression [139,140].

Why does the loss of ATRX/DAXX lead to ALT in cancer cells, but genetic knock-out of ATRX/DAXX does not? ALT appears to rely on a certain level of replication stress at telomeres to perpetuate itself. There seems to be a “Goldilocks zone” of replication stress that can maintain ALT without leading to cell toxicity. If too much replication stress occurs in ALT cells, as in the case of FANCM depletion, for example [141,142], ALT activity is increased to the point of toxicity and cells cannot survive. Remarkably, it was recently observed that BIR itself produces replication stress, creating a self-reinforcing cycle of DNA damage to perpetuate ALT telomere maintenance [122]. Thus, the normal activities of ATRX/DAXX at telomeres maintain heterochromatin silencing, leading to reduction in TERRA transcription, reduced G4s and R-loops, and less replication stress (Figure 2). In the absence of ATRX/DAXX, the levels of replication stress may increase, but not enough for the ALT mechanism to be self-sustaining. Only through additional insults does the level of replication stress attain the vicious cycle necessary to perpetuate ALT. 

## 5. Conclusions

Important questions about the role of ATRX/DAXX in protecting the genome remain. ATRX is not a canonical helicase and has not been shown to have G4 unwinding function. While it clearly is important for managing structured DNA sequences in the genome, the exact molecular mechanism remains murky. More biochemical assays, though difficult in a protein of this size, would be helpful to elucidate this question. Additionally, the proposed role of ATRX in opposition to BLM, promoting crossover resolution instead of dissolution, overturned the conventional wisdom that mammalian DNA repair results in mostly non-crossover products. Follow-up work should be done to confirm and expand this important finding. Finally, it is tantalizing to imagine that TERRA may act as a platform guiding assembly of ALT factors. In that case, the role of ATRX as a TERRA antagonist might be central to ALT suppression. It will be fascinating to learn more about this critical ncRNA.

Future work related to DAXX is likely to focus on the many other binding partners and complexes it participates in. Particularly fascinating is the notion that DAXX can act as a protein folding chaperone [101]. Considering the wildly disordered nature of ATRX, it is tempting to wonder the extent to which DAXX may be acting merely as a scaffold to deliver a folded ATRX to the proper location. Of course, DAXX also delivers H3.3/H4, but it remains unknown to what extent deposition of H3.3 *per se*, as opposed to another H3 variant, is essential to the function of ATRX/DAXX in ALT suppression. More work is necessary to understand to what extent HIRA can compensate for H3.3 deposition at telomeres.

Many questions surround the role of ATRX/DAXX mutations in tumorigenesis. It is unclear whether ATRX/DAXX mutations are early or late acquisitions in tumorigenesis, and indeed some evidence points each way. It’s likely that the answer may vary by tumor type. Are all ATRX/DAXX mutations loss of function? Though most cancer-associated ATRX/DAXX mutations are truncations, missense mutations are also observed and their significance is not well comprehended. We can only expect that a better understanding of these guardians of the genome may open new pathways for targeted therapies, and ultimately lessen the burden of disease.

## Figures and Tables

**Figure 1 genes-14-00790-f001:**
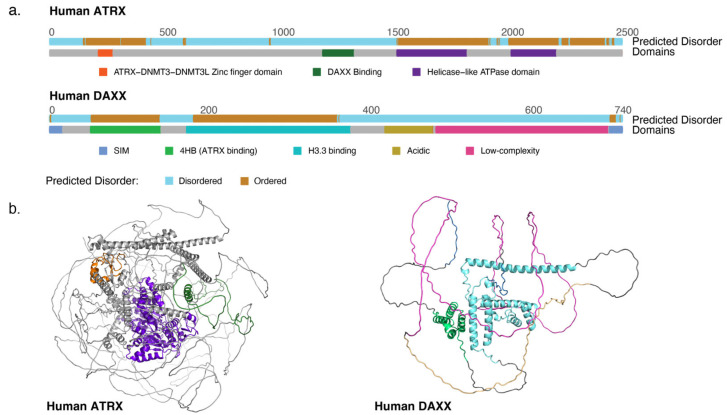
(**a**) Diagram of the ATRX and DAXX proteins outlining major domains and indicating predicted disordered regions. (**b**) AlphaFold models of ATRX and DAXX illustrating the largely disordered nature of the proteins, with domains colored as in (**a**) [23,24].

**Figure 2 genes-14-00790-f002:**
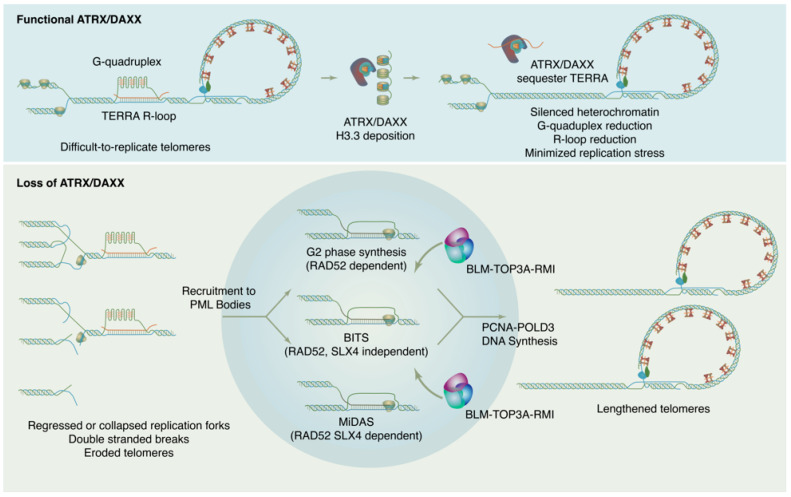
ATRX/DAXX suppresses ALT by maintaining chromatin states and reducing replication stress at telomeres. In absence of ATRX/DAXX, chromatin loses silencing, TERRA transcription increases, G4s and R-loops increase, and replication stress results. This replication ultimately feeds forward into self-perpetuating ALT.

## Data Availability

Not applicable.

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
