# Peer review of "ATRX/DAXX: Guarding the Genome against the Hazards of ALT"

_genes, 2023, doi:10.3390/genes14040790_

Round 1
Reviewer 1 Report
The review is of good quality and will be helpful to the researches involved in telomere field. The logic is comprehensible; the language is rich, but not too sophisticated. The last picture (Fig.2) is very helpful. Only two points:
1. Different protein chains are discussed. A lot of abbreviations used in the review. Abbreviations have definitions, but during travel through text it is difficult to keep in mind all of them. Reviewer do not know is it allowed by the Journal rules, but, it’ll be helpful if kind of glossary with all abbreviation definitions will be added.
2. Authors proved the importance of the non-coding RNA TERRA in the arranging gene ansambel, which suppresses ALT and maintain normal chromatin structure at telomere. ATRX/DAXX is a part of the normal complex. The underlined role of TERRA is the review advantage. Reviewer believe that in case of other structures, in which ATRX or DAXX involved other non-coding RNA (ncRNA) should be found as guides. If PML bodies do not contain RNA, that does not exclude their guiding role. NcRNA is known for theirs burst transcription (Pontis et al., 2022; doi: 10.1038/s41467-022-34800-w). It would be nice to mention such a possibility in the Conclusion.
Author Response
- Thank you for the feedback. I agree that I find a glossary section to be very helpful in a review. Unfortunately the journal format does not offer this section so far as I know.
- Thank you for this thoughtful idea. I have added a comment about the possibility of TERRA guiding ALT factor assembly to the conclusion.
Reviewer 2 Report
ATRX/DAXX is responsible for depositing H3.3 at pericentric and telomeric heterochromatin and has roles in ameliorating replication in repeat sequences and in promoting DNA repair. In this review, the authors explored the normal function of ATRX and DAXX in safeguarding the genome, and describe recent advances for the ATRX/DAXX Complex, localization and function of ATRX/DAXX, and suppression of ALT. It is valuable for understanding of how mutations in the ATRX/DAXX complex promote acquisition of the ALT phenotype.
Minor questions:
1. Line 392. Break-induced replication should be “BIR” when it was secondly used.
2. Figure 2 should be showed more explain and label. I failed to completely understand it.
Author Response
- Thank you for this helpful feedback. I have made the correction.
- Figure 2 has been updated to incorporate your feedback. I hope it is clearer.